**Data Availability Statement:** Data cannot be shared publicly because of containing potentially sensitive information by the decision of the ethics

# Demand for family planning satisfied with modern methods and its associated factors among married women of reproductive age in rural Jordan: A cross-sectional study

**Makiko Komasawa**[1]*, **Motoyuki Yuasa**[1], **Yoshihisa Shirayama**[2], **Miho Sato**[3], **Yutaka Komasawa**[4], **Malak Alouri**[5]

1 Department of Public Health, Faculty of Medicine, Juntendo University, Bunkyo-ku, Tokyo, Japan, 2 Faculty of International Liberal Arts, Juntendo University, Bunkyo-ku, Tokyo, Japan, 3 School of Tropical Medicine and Global Health, Nagasaki University, Nagasaki city, Japan, 4 Atelier 514, Setagaya-ku, Tokyo, Japan, 5 Directorate of Woman and Child health, Ministry of Health of Jordan, Director of Woman and Child Health Directorate, Amman, Jordan

* mkomasa@juntendo.ac.jp

## Abstract

### Background

A novel indicator, 'percentage of women of reproductive age who are sexually active and who have their demands for FP satisfied with modern contraceptive methods (mDFPS)', was developed in 2012 to accelerate the reduction of unmet needs of family planning (FP). In Jordan, unmet needs for modern contraception remain high. To address this situation, this study measured the mDFPS and identified its associated factors in rural Jordan.

### Methods

This cross-sectional study included married women of reproductive age (15–49 years) from ten villages in Irbid Governorate, Jordan, where advanced health facilities are difficult to reach. A two-stage stratified sampling with random sampling at the household stage was used for this field survey which was conducted between September and October 2016. Univariate analysis was used to assess the differences between mDFPS and unmet mDFPS groups. Logistic regression analysis was performed to identify the correlates of mDFPS.

### Results

Of 1019 participants, 762 were identified as needing modern contraception. mDFPS coverage accounted for 54.7%. The most significant factors associated with mDFPS were the husband's agreement on FP (adjusted odds ratio [AOR]: 15.43, 95% confidence interval [CI]: 5.26–45.25), knowledge of modern contraceptives (AOR: 8.76, 95% CI: 5.72–13.40), and lack of awareness of the high risk of conception in the postpartum period (AOR: 2.21, 95% CI: 1.41–3.47). Duration of current residence, receipt of FP counselling at health centres and number of living children were also correlated. In addition, 95.3% of local women

committee of Juntendo University. Data are available from Prof. Myo Nyein Aung (myo@juntendo.ac.jpv), Global Health Promotion Research Center, Faculty of International Liberal Arts, Juntendo University, for researchers who meet the criteria for access to confidential data.

**Funding:** This study was funded by the Japan International Cooperation Agency (JICA) (https://www.jica.go.jp/english/about/index.html). The funder had no role in study design, data collection and analysis, decision to publish, or preparation of the manuscript.

**Competing interests:** NO authors have competing interests.

were aware of the presence of health centres that were mostly located in a 10-minute walking distance.

## Conclusion

To increase mDFPS, this study suggested that accelerating male involvement in FP decision-making is necessary through community-based health education. Furthermore, expanding FP services in village health centres and improving the quality of FP counselling in public health facilities are required to correct misconceptions about modern methods among rural women.

## Introduction

An estimated 40% of pregnancies were unintended in developing countries in 2012 [1]. It has also been reported that nearly 90% of unintended pregnancies in low- and middle-income countries can be prevented by using modern contraceptive methods [2]. An international consortium, Family Planning 2020 (FP2020), was established in 2012 to accelerate modern contraceptive use to reduce unmet needs of family planning (FP) by 2020 [3], FP2020 developed a novel indicator for one of the FP goals, 'demand for family planning satisfied with modern methods (mDFPS)' [4]. mDFPS is defined as the percentage of women (or their partners) who seek to avoid or delay pregnancy but who do not use any modern contraceptive methods [5]. A major difference between mDFPS and the previous contraceptive prevalence rate (CPR) is that this new measure does not consider traditional contraceptive methods.

A few studies have examined mDFPS based on existing data from either demographic health surveys (DHS) or multiple indicator cluster surveys at national level [6, 7]. The studies mostly focused on countries with low CPRs, especially in Sub-Saharan Africa, but rarely focused on Middle Eastern countries. The first comprehensive study on mDFPS was conducted by Westoff in 2012 [6]. He estimated that mDFPS accounted for 47% of women worldwide and illustrated that in general lower education level and lower economic status leads to lower mDFPS coverage. With regard to exposure to information on FP, the study reported that FP messages on television and radio had a positive effect on mDFPS in countries with low CPRs. Another study also found that being younger, poor, having lower education or living in a rural area meant that women tended to have lower mDFPS than the rest of the population [7].

Jordan's CPR increased from 40% in 1990 to 56% in 2002, and to 61% in 2012; however, the CPR decreased to 52% according to the Jordan Population and Family Health Survey (JPFHS) in 2017–2018 [8, 9]. The Jordanian government explained that one of the major reasons behind this drop was the decline in the use of traditional contraceptive methods. Another reason may be the influx of Syrian refugees, who tended to have lower CPR than Jordanians. However, this information was not clarified because CPR data by nationality was not available in the previous JPFHS in 2012 [8]. JPFHS 2012 estimated that the national average mDFPS in 2012 was 58%, which was not high compared with that of other Middle Eastern countries, including 80% in Egypt, 73% in Morocco and 56% in Turkey [6]. Moreover, Bongaarts and Casterline [10] recently classified Jordan as a pre-fertility-transition country, one of a few such countries outside of Sub-Saharan Africa, and predicted that the country's high unplanned pregnancy rate and contraceptive failure rate would continue until the mean number of desired children decreased to 3.0 per woman.

This study measured the mDFPS based on field data in rural Jordan and identified factors associated with mDFPS.

## Materials and methods

### Study setting

This study used the data from a baseline survey conducted by a project funded by the Japan International Cooperation Agency (JICA) in 2016–2018 [11]. The purpose of the project was to strengthen the service delivery function of village health centres (VHCs) in rural villages where advanced health facilities are lacking. Before the project, most VHCs could not provide family planning services. The details of the project design can be found in our study published elsewhere [12]. The study team selected Irbid Governorate as the study site because no similar assistance from other donors had been implemented there before this project. Irbid Governorate is located in northern Jordan, 100 km from the capital Amman, with an estimated population of 1.8 million in 2017 [13]. The study target included currently married women of reproductive age (15–49 years) because only married women are culturally considered as being sexually active in Jordan. Using a structured questionnaire, trained and experienced female researchers conducted face-to-face interviews in Arabic at the individual's house. Most questions were drawn from JPFHS 2012 [8], and some questions were added from an earlier study in Jordan [14]. Data collection was carried out between September and October 2016.

### Sampling

A two-stage stratified sampling was used for the present study. The first stage involved selecting ten villages that had VHCs from four health districts in the Irbid Governorate. The five intervention VHCs were purposely selected by the project team and the Ministry of Health from villages where advanced health facilities are difficult to reach. To match each intervention village, five control villages in the respective health districts were selected by considering similar geographical and socioeconomic characteristics. The second stage involved selecting households by systematic random sampling in each village, using the household frame of the 2015 Jordan Census which was the same as the methodology used in the JPFHS 2012 [8]. The sampling allocation is shown in S1 Table of the Supporting information, and more detailed sampling procedures can be found in our previous study [12]. When a household did not contain a woman eligible for participation, the household was replaced by the nearest one. In case of more than one eligible woman in a household, the researcher randomly selected one participant.

The sample size was calculated based on the following assumptions: 50% CPR detection based on JPFHS 2012 at 95% confidence level (CI) and 80% power. The minimal sample size was then determined to be 384 [12]. In addition, assuming that 80% of the interviewed women have a need for modern contraception, the required sample size was 480 individuals. The original sample size for the purpose of the project of 1000 in total, covered our required sample size [12].

### Selected study variables

The essential outcome variable was mDFPS based on the definition of the World Health Organization (WHO) [5]. The numerator for mDFPS was the number of currently married women who were using any modern contraceptive methods in the one month period prior to the survey. Modern methods included pills, condoms (male and female), intrauterine devices, injectables, implants, diaphragms, spermicidal agents (foam/jelly), sterilisation (male and female)

[15], and the lactational amenorrhea method (LAM: exclusively breastfeeding within six months after birth), based on the JPFHS definition [8]. The denominator was the total number of women having needs for modern methods among women of reproductive age (15–49 years) who are married or in a union. It included the following:

a. all fecund women who are using any modern or traditional contraceptives in the last one month;

b. all pregnant women whose pregnancies were unintended or mistimed at the beginning of the pregnancies;

c. all postpartum amenorrhoeic women who were not using any contraceptives and whose latest birth was unintended or mistimed; and

d. all other fecund women who were not using any contraceptives, those who wanted to stop childbearing, those who wished to postpone childbearing for more than two years or those who did not have any plan for childbearing.

Twenty-two independent variables were categorised into four groups: socioeconomic factors; reproduction status; husband-related factors and exposure to reproductive health (RH) information. A variable named 'knowledge on modern methods' was regrouped from four options into two options (yes/no): only 'modern methods are more effective' was considered 'yes', and other options including 'modern methods are less effective', 'modern methods are equally effective' and 'don't know' were considered 'no'. Concerning the variable 'main decision-maker for contraceptive use', two options ('husband alone' and 'others') were combined and set as 'others' because of the small number of respondents.

## Statistical analysis

Firstly, we identified women in need of modern contraception from all respondents. Secondly, univariate analysis was conducted to assess the differences between two groups, mDFPS versus unmet mDFPS, using the chi-square or Fisher's exact tests. Of the 22 independent variables, nine were selected by univariate analysis because they were statistically significant. Subsequently, two variables in husband-related factors were excluded because one could represent the other two. Finally, seven independent variables with 'nationality' as an adjusting variable were entered into multivariate logistic regression analysis (mDFPS versus unmet mDFPS). Statistical significance was set at 0.05. SPSS version 26 (IBM, Chicago, USA) was used for the statistical analysis.

## Ethical considerations

Written informed consent was obtained from each participant after full explanation of the study purposes. For married girls aged 15–17 years, in addition to the written consents, verbal consents were obtained from their husbands or mothers-in-law before contacting them. After screening collected data, all data were anonymised. The design and implementation of the study were approved by the Ministry of Interior, Jordan (reference no. 3058/4/2/6; 7 September 2016) and the Ethics Committee of the Faculty of Medicine, Juntendo University, Japan (reference no. 2015104; 29 January 2016).

## Results

In total, 1019 women were successfully interviewed. Of these, 762 (74.8%) women had a need for modern contraceptive use according to the WHO definition [5]. The basic characteristics

of the respondents are summarised in Table 1. The mean duration of schooling of the women was more than 11 years. The mean number of living children was 3.9, and the gap between the number of desired children and the number of living children was 0.4. Regarding nationality, the majority were Jordanian (95.5%), followed by Syrian (4.2%) and other nationalities (0.3%) (S2 Table). Of all women having a need for modern contraception, 54.7% were currently using modern contraceptive methods, 26.5% were using traditional methods and 18.8% did not use any method. Popular modern methods among the users were IUDs and pills (32.4% and 11.4%, respectively); whereas, the most common traditional method was withdrawal (24.5%) (S3 Table). In terms of accessibility to health facilities, 95.3% of respondents were aware of the nearest public health facility, which was a VHC, and 78.0% have used it in the last year (S4 and S5 Tables). More than 80% of women can access the nearest VHC within 10 minutes mostly on foot (S6 and S7 Tables). Among women who have used VHCs in the last year, services frequently used were a general medical examination by a part-time general practitioner and an immunization program for children. The major reasons for non-use of VHCs were services they needed were not available (S9 Table).

The mDFPS coverage accounted for 54.7% (Table 2). Tables 2–5 show the differences between the mDFPS and unmet mDFPS groups. Table 2 describes the socioeconomic characteristics of the two groups. Remarkably, nearly half the women completed secondary education in both groups. One variable (i.e. 'duration of current residence') showed statistically significant difference between the two groups; whereas, age, education level, work experience, and household income did not.

Table 3 presents the reproductive status of the two groups. 'Age at first marriage', 'number of living children', 'knowledge on modern methods', and 'in postpartum amenorrhea' were significantly different between the mDFPS and unmet mDFPS groups. Conversely, 'number of desired children' did not show a significant difference between the two groups. Notably, 15.4% of the participants were experiencing postpartum amenorrhea at the time of survey. We also asked about the reason(s) for non-use of contraceptives to women who were currently not practising FP, and the majority of the reasons were related to temporary infertility and fertility preference (S10 Table). To disclose underlying reasons for non-use of modern contraceptives, we asked a trial question on the perception of modern methods among community people, and the most dominant answer was 'fear of health problems' accounting for 84.5% (S11 Table).

Husband-related factors are summarised in Table 4. All factors showed statistically significant differences between the two groups. The majority of women (92.9%) perceived that their

**Table 1. Basic characteristics and contraceptive use (n = 762).**

|  | n | Mean | SD |
|---|---|---|---|
| Age (years) | 762 | 35.2 | 7.6 |
| Schooling (years) | 761 | 11.5 | 3.0 |
| Age at first marriage (years) | 761 | 20.8 | 4.1 |
| Age at first delivery (years) | 743 | 22.1 | 3.8 |
| Number of living children | 761 | 3.9 | 1.9 |
| Number of desired children | 760 | 4.3 | 1.4 |
| Monthly household income (Jordan dinar) | 734 | 386.2 | 182.2 |
| Current contraceptive use (%) |  |  |  |
| All modern methods | 417 | 54.7 |  |
| All traditional methods | 202 | 26.5 |  |
| No use of any methods | 143 | 18.8 |  |

**Table 2. Socioeconomic characteristics of the two groups: Demand for family planning with modern methods (mDFPS) and unmet mDFPS (n = 762).**

| | Total n (%) | Unmet mDFPS (%) | mDFPS (%) | *p*-value |
|---|---|---|---|---|
| All | 762 (100.0) | 345 (45.3) | 417 (54.7) | |
| Age (years) | | | | 0.233 |
| <35 | 353 (46.3) | 168 (48.7) | 185 (44.4) | |
| ≥35 | 409 (53.7) | 177 (51.3) | 232 (55.6) | |
| Schooling (years) | | | | 0.097 |
| ≤10 | 212 (27.0) | 97 (28.1) | 115 (27.6) | |
| 11 | 204 (26.8) | 80 (23.2) | 124 (29.8) | |
| ≥12 | 345 (45.3) | 168 (48.7) | 177 (42.5) | |
| Missing | 1 | 0 | 1 | |
| Work experience in the last year | | | | 0.438 |
| No | 87 (11.4) | 36 (10.4) | 51 (12.2) | |
| Yes | 675 (88.6) | 309 (89.6) | 366 (87.8) | |
| Duration of current residence (years) | | | | <0.001 |
| ≤2 | 156 (20.5) | 92 (26.7) | 64 (15.3) | |
| >2 | 605 (79.5) | 252 (73.3) | 353 (84.7) | |
| Missing | 1 | 1 | 0 | |
| Monthly household income (Jordan dinars) | | | | 0.117 |
| <350 | 306 (41.7) | 148 (44.8) | 158 (39.1) | |
| ≥350 | 428 (58.3) | 182 (55.2) | 246 (69.9) | |
| Missing | 28 | | | |

husbands agreed to contraceptive use; nevertheless, there was a gap of 13.5% between the groups (99.0% for mDFPS and 85.5% for unmet mDFPS). The main decision-makers for contraceptive use among married couples were 'wife and husband' jointly (88.6%), whereas, 'others' including 'husband alone' and 'parents', accounted for only 4.2%. Nearly one quarter of the women perceived that their husbands wanted more children than they did.

Trends of exposure to RH information by source are listed in Table 5. With respect to person-to-person communication channels (i.e. counselling at health centres, private clinics, and non-government organisations), counselling at private clinics was the most common exposure channel for RH information. Approximately 70% of women reported exposure via television, followed by the Internet. Among RH information channels, only counselling at health centres showed a significant difference between the two groups.

Table 6 shows the result of multivariate logistic regression analysis of factors associated with mDFPS (reference: unmet mDFPS group). Six variables were associated with mDFPS: 'husband's agreement on contraceptive use' (adjusted odds ratio [AOR]: 15.43, 95% CI: 5.26–45.25); 'knowledge on modern methods' (AOR: 8.76, 95% CI: 5.72–13.40); 'in postpartum amenorrhea' (AOR: 2.21, 95% CI: 1.41–3.47); 'duration of current residence' (AOR: 1.83, 95% CI: 1.19–2.83); 'number of living children' (AOR: 1.59, 95% CI: 1.12–2.27); and 'counselling at health centres' (AOR: 1.66, 95% CI: 1.13–2.44).

## Discussion

This study measured mDFPS coverage in rural Jordan based on data collected from the field. The mDFPS coverage (54.7%) in this study was similar to the estimated mDFPS in rural Jordan in 2017–2018 (53.4%) [9]. Major factors associated with mDFPS in previous studies across the world (i.e. sociodemographic factors) were not detected in the present study [2, 6]. However, the major factors related to unmet mDFPS in our study were similar to widespread

**Table 3. Reproductive status of the two groups: Demand for family planning with modern methods (mDFPS) and unmet mDFPS (n = 762).**

| | Total n (%) | Unmet mDFPS (%) | mDFPS (%) | p-value |
|---|---|---|---|---|
| Age at first marriage | | | | 0.032 |
| <20 | 313 (41.1) | 127 (36.9) | 186 (44.6) | |
| ≥20 | 448 (58.9) | 217 (63.1) | 231 (55.4) | |
| Missing | 1 | 1 | 0 | |
| Number of living children | | | | <0.001 |
| 0–3 | 325 (42.7) | 175 (50.9) | 150 (36.0) | |
| 4–5 | 280 (36.8) | 103 (29.9) | 177 (42.4) | |
| ≥6 | 156 (20.5) | 66 (19.2) | 90 (21.6) | |
| Missing | 1 | 1 | 0 | |
| Number of living children | | | | 0.500 |
| 0–3 | 159 (20.9) | 78 (22.7) | 81 (19.4) | |
| 4–5 | 464 (61.1) | 203 (59.2) | 261 (62.6) | |
| ≥6 | 137 (18.0) | 62 (18.1) | 75 (18.0) | |
| Missing | 2 | 2 | 0 | |
| Knowledge on modern methods | | | | <0.001 |
| Yes | 570 (74.8) | 190 (55.1) | 380 (91.1) | |
| No | 192 (25.2) | 155 (44.9) | 37 (8.9) | |
| In postpartum amenorrhea | | | | 0.002 |
| Yes | 117 (15.4) | 68 (19.7) | 49 (11.8) | |
| No | 645 (84.6) | 277 (80.3) | 368 (88.2) | |

factors related to unmet needs for FP reported by past studies, which are discussed individually in the following paragraphs.

The most apparent factor associated with mDFPS in our analysis was the spousal-related factor. Particularly, 'husband's agreement on contraceptive use' was a dominant factor associated with mDFPS. Univariate analysis revealed that the husband's equal participation in decision-making for FP use and fertility preference also affected mDFPS. Numerous studies on unmet needs for FP have reported that the husband's fertility preference and his attitude towards FP were crucial factors influencing women's contraceptive use [16–26]. In Egypt, for

**Table 4. Husband-related factors of the two groups: Demand for family planning with modern methods (mDFPS) and unmet mDFPS (n = 762).**

| | Total n (%) | Unmet mDFPS (%) | mDFPS (%) | p-value |
|---|---|---|---|---|
| Husband's agreement on contraceptive use | | | | <0.001 [a] |
| Yes | 708 (92.9) | 295 (85.5) | 413 (99.0) | |
| No | 54 (7.1) | 50 (14.5) | 4 (1.0) | |
| Main decision-maker for contraceptive use | | | | <0.001 |
| Wife and husband | 675 (88.6) | 291 (84.3) | 384 (92.1) | |
| Wife only | 55 (7.2) | 27 (7.8) | 5 (6.7) | |
| Other | 32 (4.2) | 27 (7.8) | 28 (1.2) | |
| Husband's fertility preference | | | | 0.018 [a] |
| Same | 533 (69.9) | 242 (70.1) | 291 (69.8) | |
| More than wife | 168 (22.0) | 68 (19.7) | 100 (24.0) | |
| Less than wife | 48 (6.3) | 24 (7.0) | 24 (5.8) | |
| Don't know | 13 (1.7) | 11 (3.2) | 2 (0.5) | |

[a] Fisher's exact test.

**Table 5. Exposure to reproductive health information of the two groups: Demand for family planning with modern methods (mDFPS) and unmet mDFPS (n = 762).**

|  | Total n (%) | Unmet mDFPS (%) | mDFPS (%) | *p*-value |
|---|---|---|---|---|
| Counselling at health centres [a] |  |  |  | 0.001 |
| Yes | 214 (28.1) | 77 (22.3) | 137 (32.9) |  |
| No | 548 (71.9) | 268 (77.7) | 280 (67.1) |  |
| Counselling at private clinics |  |  |  | 0.366 |
| Yes | 325 (42.7) | 141 (40.9) | 184 (44.1) |  |
| No | 437 (57.3) | 204 (59.1) | 233 (55.9) |  |
| Counselling at non-governmental organisations |  |  |  | 0.931 |
| Yes | 67 (8.8) | 30 (8.7) | 37 (8.9) |  |
| No | 695 (91.2) | 315 (91.3) | 380 (91.1) |  |
| Group lecture in community |  |  |  | 0.430 |
| Yes | 36 (4.7) | 14 (4.1) | 22 (5.3) |  |
| No | 726 (95.3) | 331 (95.9) | 395 (94.7) |  |
| Television |  |  |  | 0.885 |
| Yes | 517 (67.8) | 235 (68.1) | 282 (67.6) |  |
| No | 245 (32.2) | 110 (31.9) | 135 (32.4) |  |
| Printed material |  |  |  | 0.209 |
| Yes | 191 (25.1) | 79 (22.9) | 112 (26.9) |  |
| No | 571 (74.9) | 266 (77.1) | 305 (73.1) |  |
| Internet |  |  |  | 0.757 |
| Yes | 276 (36.2) | 127 (36.8) | 149 (35.7) |  |
| No | 486 (63.8) | 218 (63.2) | 268 (64.3) |  |
| SMS text |  |  |  | 0.589 |
| Yes | 32 (4.2) | 13 (3.8) | 19 (4.6) |  |
| No | 730 (95.8) | 332 (96.2) | 398 (95.4) |  |
| Relative/family |  |  |  | 0.292 |
| Yes | 170 (22.3) | 83 (24.1) | 87 (20.9) |  |
| No | 592 (77.7) | 262 (75.9) | 330 (79.1) |  |

[a] Health centres include primary health centres and comprehensive health centres.

example, most users (91.6%) perceived that their husbands agreed on contraceptive use; whereas, approximately only one-fourth of non-users (26.4%) felt that their husbands agreed [17]. Many studies suggested that FP programmes must mobilise husbands and other influential male members in families and communities, as a whole, in order to address the negative social norms or barriers towards use of modern methods in rural settings [19, 21, 25, 27, 28]. Nevertheless, most of these studies, including our study, relied on data by asking about women's perceptions of their husbands' acceptance of FP use. Casterline et al. [18] suggested that there were considerable discrepancies between the wife's perceptions of her husband's FP acceptance and his actual acceptance. In-depth research of both sexes of the couples could investigate the real preferences and attitudes towards modern methods in rural Jordan.

The other important factor was accurate knowledge of the effectiveness of modern contraceptives. Women who perceived the effectiveness of modern methods were nearly nine times more likely to use modern contraceptives than those in the other group. The latest review studies identified that remaining obstacles to non-use of contraception in countries with an increase the CPR and rising education levels were fears of side-effects, adverse health risks, and the risk of infertility [25, 29–32]. Although we asked women who were currently not

**Table 6. Factors associated with demand for family planning with modern methods (mDFPS) (n = 759).**

| | Adjusted Odds Ratio [a] (95% CI) [b] | p-value |
|---|---|---|
| Age at first marriage (years) | | |
| <20 | 1 | 0.311 |
| ≥20 | 1.20 (0.84–1.71) | |
| Number of living children | | |
| ≤3 | 1 | 0.010 |
| >3 | 1.59 (1.12–2.27) | |
| Knowledge on modern methods | | |
| Yes | 8.76 (5.72–13.40) | |
| No | 1 | <0.001 |
| Duration of current residence (years) | | |
| ≤2 | 1 | 0.006 |
| >2 | 1.83 (1.19–2.83) | |
| In postpartum amenorrhea | | |
| Yes | 1 | 0.001 |
| No | 2.21 (1.41–3.47) | |
| Husband's agreement on contraceptive use | | |
| Yes | 15.43 (5.26–45.25) | |
| No | 1 | <0.001 |
| Counselling at health centres | | |
| Yes | 1.66 (1.13–2.44) | |
| No | 1 | 0.011 |

Excluding 1 missing data from 'age at first marriage', 'knowledge on modern methods', and 'duration of current residence".

[a] Adjusted with nationality.

[b] CI: confidence interval.

practising FP for their reason(s) for non-use of contraceptives, no insightful results appeared (S10 Table). On the contrary, in response to our question on the reasons for non-use of modern methods among people in the community, 'fear of health problems' was the most dominant reason (S11 Table). This indicated that a vague sense of fear of health effects may be the underlying reason for non-use of modern methods in rural Jordan. A prior study in Jordan also reported that a latent fear of health risks associated with hormonal contraceptive methods and IUDs might have turned into a perception of modern methods being ineffective [14]. These findings implied that the rural population in Jordan fears potential side-effects and adverse health effects as well as the risk of infertility. Campbell et al. [33] urged that misinformation may be the real barrier to using modern methods. Overall, our analysis suggested that adequate education regarding modern methods should be enhanced by providing high-quality person-to-person counselling at public health centres in rural Jordan to remove all misperceptions of modern methods by providing high-quality person-to-person counselling at public health centres in rural Jordan [31, 34, 35].

This study also revealed that a low perception of risk of getting pregnant during the postpartum period affected mDFPS. Women experiencing postpartum amenorrhea were more than twice as likely to have unmet mDFPS than were the other group. Previous studies pointed out that many women were not aware of the high risk of pregnancy during the postpartum period [26, 30, 31, 36]. A study from the state of Virginia in the USA reported that mothers who attended their postpartum care visit were 1.44 times more likely to start FP in the

postpartum period than those who did not attend [37]. In 2016, the Jordanian Ministry of Health introduced a new regulation to the effect that mothers who do not practice LAM should start contraception after 21 days of delivery to avoid unintended pregnancies [38]. Our results, however, showed that this regulation had not yet reached rural communities at the time of the survey. Because integrating FP counselling into antenatal/postnatal care is effective [2, 21], the ministry needs to facilitate the provision of FP counselling during antenatal/postpartum care visits at health facilities and provide appropriate contraceptives to mothers with contraceptive needs at postnatal visits as a part of the continuum of care for maternal and child health [2, 20, 21].

We found that local women were aware of and could easily access VHCs in rural Jordan. Meanwhile, comprehensive or primary health centres were the most popular places where modern methods could be obtained (S12 Table). This implies that the VHC services were limited and did not meet women's needs, especially with regards to FP services. In terms of information on RH only counselling at public health centres had a positive influence on mDFPS. Counselling at private clinics was not associated with mDFPS, even though the exposure rate was higher (42.7%) than that of health centres (28.1%) (Table 5). Earlier studies in Jordan reported that women, especially those in urban areas, depended highly on private clinics for modern contraception [27, 39, 40]. Campbell et al. [33] determined a close correlation between travel time to a health facility and women's use of contraceptives. Overall, our findings highlighted the important role of public health centres to increase mDFPS coverage among married women in rural Jordan. In this respect, expanding FP services at VHCs, and increasing people's awareness of the usefulness and safety of modern contraceptives may be a key strategy for accelerating mDFPS in rural Jordan.

Our analysis showed that women living in the current community for more than two years were 1.83 times more likely to have their demands for modern methods met than women living in the current community for a shorter period. Earlier studies in Kenya and Zambia presented that migrant women in rural areas did not show significant effects on using modern methods compared with non-migrant women in rural areas [41, 42]. Our result, however, may be interpreted as meaning that longer residence generates a competency to access available health resources in communities, subsequently leading to use of their services.

With respect to the migrant issue, we originally considered Syrian refugees and other displaced populations in Jordan; however, only few Syrians and other nationalities were living in our study areas. It was because 83% of Syrian refugees in Jordan live in urban areas [43] where they can easily access refugee services with less travel cost and obtain necessary information in a close-knit community. Further studies are required to explore the Syrian refugees' situations in terms of RH and to establish a resilient health system for all.

In our analysis, the number of living children was also associated with mDFPS, in line with the results of earlier studies [22, 44]. Women with four or more children were more likely (1.59 times) to use modern contraceptive methods than those with fewer than four children. This may be explained by the stronger motivation to avoid pregnancies after reaching the desired number of children. Conversely, there was no association between the number of desired children and mDFPS. Many studies reported that women's preferences on childbearing in terms of numbers and timing were ambivalent and their desired number of children was changing over their lifetime [22, 23, 35]. Additionally, mothers may hesitate to respond with a number smaller than their actual number of living children or they may accept an unplanned pregnancy after the birth of the child [22, 23, 26, 35]. To examine these phenomena, further psychological approaches are required.

The strength of this study (to the best of our knowledge) was that it is the first community-based study in Jordan on mDFPS with adequate power to identify the mDFPS coverage.

However, there are several limitations. The most significant limitation of this study may be possibility of containing biases from self-reporting by local women, such as practicing either modern or traditional contraception, fertility preferences, or the wantedness of recent births/ pregnancies. Some bias may have been included; therefore, mDFPs may have been underestimated [35]. Secondly, this study was conducted only in the Irbid Governorate and could not represent the whole of Jordan. In addition, considering the current situation in Jordan, the influence of Syrian refugees cannot be ignored; however, our study could not examine this aspect due to the small sample size. Thirdly, as mentioned earlier, using women's perceptions of husband's preference generates some unreliability of reality because of discrepancies between the parties. Fourthly, owing to our limited sample size, this study did not determine the factors related to birth spacing and birth limitation, which are important parameters for identifying further needs regarding mDFPS [25, 26]. Nevertheless, our study began to reveal the current status of mDFPS in evidence and suggested the need for further strengthening of FP programmes in rural Jordan.

## Conclusion

The current mDFPS coverage was still almost half in rural Jordan. Our analysis highlighted that significant factors associated with mDFPS were spousal agreement for FP use, awareness of the effectiveness and safety of modern contraceptives, and lack of risk of conception during the postpartum period. To increase mDFPS, our results suggested accelerating male involvement in FP decision-making is necessary through community-based health education. Furthermore, expanding FP services at VHCs and improving FP counselling in all primary health facilities, with a special focus on allaying the fear of adverse health effects from modern methods and increasing awareness on the importance of postpartum contraception are required.

## Supporting information

**S1 Table. Number of samples by village.**
(DOCX)

**S2 Table. Participants according to nationality.**
(DOCX)

**S3 Table. Contraceptive methods among women having needs for modern contraception.**
(DOCX)

**S4 Table. Awareness of the nearest village health centres.**
(DOCX)

**S5 Table. Use of village health centre in the last year.**
(DOCX)

**S6 Table. Transportation mean to the nearest village health centre.**
(DOCX)

**S7 Table. Time required to reach the nearest village health centre.**
(DOCX)

**S8 Table. Services used at village health centre.**
(DOCX)

**S9 Table. Reasons for non-use of village health centre.**
(DOCX)

**S10 Table. Reasons for non-use of family planning.**
(DOCX)

**S11 Table. Perceived reasons for non-use of modern contraceptives in community.**
(DOCX)

**S12 Table. Places for obtaining contraceptives among women currently using family planning.**
(DOCX)

## Acknowledgments

The authors would like to thank the Ministry of Health for their entire support. We also gratefully acknowledge the Department of Statistics of Jordan that assisted us in conducting the field survey. We would also like to thank the women who participated in this study.

## Author Contributions

**Conceptualization:** Makiko Komasawa, Motoyuki Yuasa, Malak Alouri.

**Data curation:** Makiko Komasawa, Yoshihisa Shirayama, Yutaka Komasawa.

**Funding acquisition:** Makiko Komasawa.

**Methodology:** Makiko Komasawa, Miho Sato.

**Project administration:** Malak Alouri.

**Supervision:** Motoyuki Yuasa.

**Validation:** Yoshihisa Shirayama, Yutaka Komasawa.

**Writing – original draft:** Makiko Komasawa.

**Writing – review & editing:** Makiko Komasawa, Motoyuki Yuasa, Yoshihisa Shirayama, Miho Sato.

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
