## [Decision Letter · Decision Letter 0]

24 Oct 2019

PONE-D-19-25354

Demand for family planning satisfied with modern methods and its determinants among married women of reproductive age in rural Jordan: A community-based cross-sectional study

PLOS ONE

Dear Makiko Komasawa,

Thank you for submitting your manuscript to PLOS ONE. After careful consideration, we feel that it has merit but does not fully meet PLOS ONE’s publication criteria as it currently stands. Therefore, we invite you to submit a revised version of the manuscript that addresses the points raised during the review process.

The first reviewer has made several comments that will help guide the process.

We would appreciate receiving your revised manuscript by 23rd November 2019. To enhance the reproducibility of your results, we recommend that if applicable you deposit your laboratory protocols in protocols.io, where a protocol can be assigned its own identifier (DOI) such that it can be cited independently in the future. For instructions see: http://journals.plos.org/plosone/s/submission-guidelines#loc-laboratory-protocols

We look forward to receiving your revised manuscript.

Kind regards,

Mary Hamer Hodges

Academic Editor

PLOS ONE

Journal Requirements:

2. During our internal checks, the in-house editorial staff noted that you conducted research or obtained samples in another country. Please check the relevant national regulations and laws applying to foreign researchers and state whether you obtained the required permits and approvals. Please address this in your ethics statement in both the manuscript and submission information.

No.

a) Please provide an amended Funding Statement that declares *all* the funding or sources of support received during this specific study (whether external or internal to your organization) as detailed online in our guide for authors at http://journals.plos.org/plosone/s/submit-now.  

b) Please state what role the funders took in the study.  If any authors received a salary from any of your funders, please state which authors and which funder. If the funders had no role, please state: "The funders had no role in study design, data collection and analysis, decision to publish, or preparation of the manuscript."

Reviewers' comments:

Reviewer's Responses to Questions

**Comments to the Author**

1. Is the manuscript technically sound, and do the data support the conclusions?

Reviewer #1: Partly

Reviewer #2: Yes

2. Has the statistical analysis been performed appropriately and rigorously? 

Reviewer #1: Yes

Reviewer #2: Yes

3. Have the authors made all data underlying the findings in their manuscript fully available?

Reviewer #1: No

Reviewer #2: No

4. Is the manuscript presented in an intelligible fashion and written in standard English?

Reviewer #1: No

Reviewer #2: Yes

5. Review Comments to the Author

Reviewer #1: This manuscript presents results from a cross-sectional survey assessing prevalence and correlates of modern contraceptive method use among married women in Irbid governorate. While some of the findings are noteworthy for adjusting program and counseling approaches to optimize modern method use, there are some concerns with the methodological approaches. A major revision is needed with better justification for and acknowledgment of bias associated including women who report recent undesired or mistimed pregnancy during or after said pregnancy, whether LAM was included as a modern method, and more thoughtful consideration of some of the findings. Overall the manuscript would benefit from review by a native English speaker or professional editor as there are numerous examples of awkward phrasing.

Abstract:

• The abstract gives the impression that mDFPS was developed solely for Jordan. Suggest reversing the order this is mentioned with the statement regarding Jordan’s rates of unmet FP need.

• The word “determinants” is inappropriate for a cross-sectional study as no causality may be attributed between the variable and the use or non-use of modern methods. Please use the word “correlates”.

• In the Methods, please define if the sample was a probabilistic (was mapping and random selection used?) or a convenience sample. Was this a household level sample or was enrollment conducted elsewhere?

• In the Conclusions section, I am concerned that improving counseling quality at public health centers is seen as the solution to increasing modern contraceptive use. After reading the Methods section within the manuscript, the household sample (which needs to be mentioned in this abstract) likely included women who are not able to access health services, potentially due to being displaced or to gender-based social norms that reduce women’s agency in accessing health care. Changing the quality of counseling at public facilities will not solve access problems but creates a more compelling reason for male engagement and also needs to consider that recently displaced populations likely don’t know where to access health care or may not have access to national public systems as non-citizens (either perceived or actual). The significant association of greater odds of using a modern method with residing in the area for more than two years argues that displaced populations may face barriers to accessing facilities in Jordan and thus cannot access modern methods or related counseling.

Introduction:

• The Introduction is generally well-written but would benefit from some alternative hypotheses for why CPR has fallen in Jordan. It is correct that mDFPS would be lower than CPR with removal of traditional methods but the dynamic population shifts in Jordan due to conflict in Iraq and Syria bear consideration that new population groups with different norms surrounding reproductive decisions should be considered.

Methods:

• Please provide further information about the setting – why was Irbid Governorate chosen? Is CPR there lower than the rest of the country? Have there been dynamic populations shifts? Please provide greater context.

• Please provide greater detail as to how the districts were chosen – this appears to be purposive sampling based on ability to access health services. Is this correct? Was the second sampling stage inclusive of community mapping to ensure a random sample?

• While other papers using mDFPS have included pregnant women reporting mistimed or unwanted pregnancies as women with contraceptive need, including this group may lead to substantive reporting bias, particularly for women who have already delivered and are amenorrheic as they would be likely to under-report that the pregnancy was unwanted or mistimed with a new infant. This is particularly as this variable was associated with modern method use.

• Regarding the postpartum group, was Lactational Amenorrhea Method queried and included with modern method use? As this group comprised a relatively large portion of surveyed women, it is important to know whether this group was included in mDFPS users if they cited LAM as their method.

• Please describe how missing data were handled in analysis.

Results:

• In the Tables, please remove the asterisks since the p-values are reported.

• Exposure to RH information by source appears to be in proportions, not by mean. Please revise the description.

Discussion:

• Earlier comments regarding query about LAM and its inclusion as a method for amenorrheic postpartum women (were these women asked if they were exclusively breastfeeding?), relying on report for whether pregnancies were unintended or mistimed, and whether women are able to access facilities vs. recommending improving counseling at facilities should be used to edit this section accordingly.

• The limitations section is fairly sparse and should first and foremost mention that all data was by participant report and is subject to recall and disclosure bias, which may be higher given the sensitivity of the topic.

• There is very little mention of length of residence in the area and possible association with refugee communities that have different norms around FP use. This information should feature prominently in this paper.

• Lines 251-253 need references to support the statement.

• The text in lines 274-277 needs to be clarified.

• The information on perceived reasons for not using contraceptives among women in their community belongs in the Results section – please do not introduce data for the first time in the Discussion section.

• Please see comments in the Abstract conclusion section to revise the Conclusion section here.

Reviewer #2: This cross sectional study aimed to assess demand for family planning satisfied with modern methods and its determinants among married women of reproductive age in rural Jordan. In general it is straightforward analysis of data and perhaps might be interesting by those who have close interest on the topic.

The only comment to imporve the paper is to better describe who are unmet mDFPS? At present inadequate information are presented.

6. PLOS authors have the option to publish the peer review history of their article (what does this mean?). If published, this will include your full peer review and any attached files.

Reviewer #1: Yes: Catherine S. Todd

Reviewer #2: No

---

## [Author Response · Author response to Decision Letter 0]

11 Dec 2019

Please kindly refer the Response to Reviewers and Response to Editors files.

---

## [Decision Letter · Decision Letter 1]

7 Jan 2020

PONE-D-19-25354R1

Demand for family planning satisfied with modern methods and its associated factors among married women of reproductive age in rural Jordan: A cross-sectional study

PLOS ONE

Dear Makiko Komasawa,

Thank you for submitting your manuscript to PLOS ONE. After careful consideration, we feel that it has merit but does not fully meet PLOS ONE’s publication criteria as it currently stands. Therefore, we invite you to submit a revised version of the manuscript that addresses the points raised during the review process.

We would appreciate receiving your revised manuscript by 6th February com/pone/ and select the 'Submissions Needing Revision' folder to locate your manuscript file.

To enhance the reproducibility of your results, we recommend that if applicable you deposit your laboratory protocols in protocols.io, where a protocol can be assigned its own identifier (DOI) such that it can be cited independently in the future. For instructions see: http://journals.plos.org/plosone/s/submission-guidelines#loc-laboratory-protocols

We look forward to receiving your revised manuscript.

Kind regards,

Mary Hamer Hodges

Academic Editor

PLOS ONE

Additional Editor Comments (if provided):

We recommend a general review of the presentation by a native English speaker familiar with the subject.

Reviewers' comments:

Reviewer's Responses to Questions

**Comments to the Author**

1. If the authors have adequately addressed your comments raised in a previous round of review and you feel that this manuscript is now acceptable for publication, you may indicate that here to bypass the “Comments to the Author” section, enter your conflict of interest statement in the “Confidential to Editor” section, and submit your "Accept" recommendation.

Reviewer #1: (No Response)

Reviewer #2: All comments have been addressed

2. Is the manuscript technically sound, and do the data support the conclusions?

Reviewer #1: Yes

Reviewer #2: Yes

3. Has the statistical analysis been performed appropriately and rigorously? 

Reviewer #1: Yes

Reviewer #2: Yes

4. Have the authors made all data underlying the findings in their manuscript fully available?

Reviewer #1: No

Reviewer #2: No

5. Is the manuscript presented in an intelligible fashion and written in standard English?

Reviewer #1: No

Reviewer #2: Yes

6. Review Comments to the Author

Reviewer #1: This revised manuscript has largely addressed prior critiques but would still benefit from some final polishing and clarification before acceptance. The manuscript also need further review by a native English speaker as there are several areas with subject-verb disagreement and other grammatical errors or awkward phrasing.

Abstract:

• As an example of awkward phrasing and need for editing, the sentence, “A filed survey, through two-stage stratified sampling of village and household levels and systematic random sampling for households, was conducted from September to October 2016.” would benefit from being revised to “Two-stage stratified sampling with random sampling at household stage was used for this field survey conducted between September and October 2016.”

• In the Results, the statement “In addition, local women were aware of and easily access public health centres in their areas.” needs proportions to put the data into perspective for this sample.

Introduction:

• Though the authors mention that a professional editor reviewed this revised manuscript, there are still multiple misspellings, issues with subject-verb agreement, and awkward phrasing (example: “”

• The statement,”Generally, low education level and low economic status cause lower mDFPS coverage” needs to be revised and referenced as low education and economic status are not causal factors of low modern FP use but are correlates.

Methods:

• This section has markedly improved. However, please capitalize proper nouns like Irbid Governorate and discuss the setting in present verb tense (current phrasing states “Irbid governorate was located in northern Jordan” – isn’t Irbid Governorate still located in northern Jordan?).

• Please add detail to specify whether households were mapped to enable random selection and what was done if a selected household did not contain an eligible woman for participation. While an earlier manuscript is referenced, it would be helpful to readers here to be able to see in the text that this is truly a probabilistic sample.

Results:

• No comments.

Discussion:

• The Limitations section still needs attention. The dependent variable, whether a woman was using a method, relies on participant report and this needs to be acknowledged as a limitation.

Reviewer #2: This is the second time that I review this manuscript. I feel the manuscript improved greatly. My comment is attended. No further comments.

7. PLOS authors have the option to publish the peer review history of their article (what does this mean?). If published, this will include your full peer review and any attached files.

Reviewer #1: Yes: Catherine Todd

Reviewer #2: Yes: Prof. Ali Montazeri

---

## [Author Response · Author response to Decision Letter 1]

25 Jan 2020

Please kindly fine the response letter dated on Jan.25

---

## [Decision Letter · Decision Letter 2]

31 Jan 2020

PONE-D-19-25354R2

Demand for family planning satisfied with modern methods and its associated factors among married women of reproductive age in rural Jordan: A cross-sectional study

PLOS ONE

Dear Makiko Komasaw,

Thank you for submitting your manuscript to PLOS ONE. After careful consideration, we feel that it has merit but does not fully meet PLOS ONE’s publication criteria as it currently stands. Therefore, we invite you to submit a revised version of the manuscript that addresses the points raised during the review process.

Ar recommended please review carefully to ensure the language properly reflects the findings as identified by the reviewer.  This would be best performed in consultation with a native English speaker.

We would appreciate receiving your revised manuscript by 15th February. To enhance the reproducibility of your results, we recommend that if applicable you deposit your laboratory protocols in protocols.io, where a protocol can be assigned its own identifier (DOI) such that it can be cited independently in the future. For instructions see: http://journals.plos.org/plosone/s/submission-guidelines#loc-laboratory-protocols

We look forward to receiving your revised manuscript.

Kind regards,

Mary Hamer Hodges

Academic Editor

PLOS ONE

Reviewers' comments:

Reviewer's Responses to Questions

**Comments to the Author**

1. If the authors have adequately addressed your comments raised in a previous round of review and you feel that this manuscript is now acceptable for publication, you may indicate that here to bypass the “Comments to the Author” section, enter your conflict of interest statement in the “Confidential to Editor” section, and submit your "Accept" recommendation.

Reviewer #1: (No Response)

2. Is the manuscript technically sound, and do the data support the conclusions?

Reviewer #1: Yes

3. Has the statistical analysis been performed appropriately and rigorously? 

Reviewer #1: Yes

4. Have the authors made all data underlying the findings in their manuscript fully available?

Reviewer #1: No

5. Is the manuscript presented in an intelligible fashion and written in standard English?

Reviewer #1: Yes

6. Review Comments to the Author

Reviewer #1: The authors have responded to all comments. However, the response may have been done in haste as there are still seeming inaccuracies that should be remedied by careful review. For example, the correction regarding Westoff et al.'s findings regarding modern contraceptive method prevalence and "that in general mDFPS is negatively associated with wealth and education." Shouldn't this be positively associated as women from higher wealth quintiles and with higher education levels are MORE likely to use modern FP methods? While the manuscript is essentially acceptable, I would urge the authors to again carefully review their text for accuracy prior to copy-editing.

7. PLOS authors have the option to publish the peer review history of their article (what does this mean?). If published, this will include your full peer review and any attached files.

Reviewer #1: Yes: Catherine Todd

---

## [Author Response · Author response to Decision Letter 2]

1 Feb 2020

Please kindly find the file named "Response to Reviewers" dated on Feb 2.

---

## [Editor Report · Decision Letter 3]

20 Feb 2020

PONE-D-19-25354R3

Demand for family planning satisfied with modern methods and its associated factors among married women of reproductive age in rural Jordan: A cross-sectional study

PLOS ONE

Dear Makiko Komasawa,

Thank you for submitting your manuscript to PLOS ONE. After careful consideration, we feel that it has merit but does not fully meet PLOS ONE’s publication criteria as it currently stands. Therefore, we invite you to submit a revised version of the manuscript that addresses the points raised during the review process.

We would appreciate receiving your revised manuscript by 29th February 2020. To enhance the reproducibility of your results, we recommend that if applicable you deposit your laboratory protocols in protocols.io, where a protocol can be assigned its own identifier (DOI) such that it can be cited independently in the future. For instructions see: http://journals.plos.org/plosone/s/submission-guidelines#loc-laboratory-protocols

We look forward to receiving your revised manuscript.

Kind regards,

Mary Hamer Hodges

Academic Editor

PLOS ONE

Additional Editor Comments (if provided):

There has been a marked improvement and is eaier to read and understand. However between the orginal version and R1 there has been a change of p value in Table 2 for age: previously reported as 0.012 and now reported as 0.233. However, the narrative is still saying 'Two variables (i.e. age and duration of current residence) showed statistically significant differences between the two groups, whereas years of schooling, work experience, and household income did not.

Please reconfirm the p value for age or adjust the narrative.

Spacing in Tables 3, and 6 after the symbol for less than or more than should be deleted.

Please harmonize decimal places for were then 1.8 times compared to (1.60 times)

---

## [Author Response · Author response to Decision Letter 3]

22 Feb 2020

Please kindly find the file named "Response to reviewers" dated on Feb 22.

---

## [Editor Report · Decision Letter 4]

2 Mar 2020

Demand for family planning satisfied with modern methods and its associated factors among married women of reproductive age in rural Jordan: A cross-sectional study

PONE-D-19-25354R4

Dear Dr.Makiko Komasawa,

We are pleased to inform you that your manuscript has been judged scientifically suitable for publication and will be formally accepted for publication once it complies with all outstanding technical requirements.

With kind regards,

Mary Hamer Hodges

Academic Editor

PLOS ONE

Additional Editor Comments (optional):

Thank you for these changes
---

## [Editor Report · Acceptance letter]

5 Mar 2020

PONE-D-19-25354R4 

Demand for family planning satisfied with modern methods and its associated factors among married women of reproductive age in rural Jordan: A cross-sectional study 

Dear Dr. Komasawa:

I am pleased to inform you that your manuscript has been deemed suitable for publication in PLOS ONE. Congratulations! Your manuscript is now with our production department. 

With kind regards,

on behalf of

Dr. Mary Hamer Hodges 

Academic Editor

PLOS ONE